# Fungicidal Efficacy of Drying Plant Oils in Green Beans against Bean Rust (*Uromyces appendiculatus*)

**DOI:** 10.3390/plants10010143

**Published:** 2021-01-12

**Authors:** Vera Breiing, Jennifer Hillmer, Christina Schmidt, Michael Petry, Brigitte Behrends, Ulrike Steiner, Thorsten Kraska, Ralf Pude

**Affiliations:** 1Institute of Crop Science and Resource Conservation, Faculty of Agriculture, INRES-Renewable Resources, University of Bonn, 53359 Rheinbach, Germany; breiing@uni-bonn.de (V.B.); jennifer.hillmer91@gmail.com (J.H.); christina.smdt@gmail.com (C.S.); r.pude@uni-bonn.de (R.P.); 2PETRYmade Oberflächentechnik, 53340 Meckenheim, Germany; info@petrymade.de; 3Marena Ltd., 26441 Jever, Germany; brigitte.behrends@marenaltd.com; 4Institute of Crop Science and Resource Conservation, Faculty of Agriculture, INRES-Plant Pathology, University of Bonn, 53115 Bonn, Germany; u.steiner@uni-bonn.de; 5Faculty of Agriculture, Field Lab Campus Klein-Altendorf, University of Bonn, 53359 Rheinbach, Germany

**Keywords:** drying plant oils, glyceride plant oils, biorationals, linseed oil, tung oil, *Uromyces appendiculatus*, bean rust, disease control, natural products

## Abstract

As biorationals, plant oils offer numerous advantages such as being natural products, with low ecotoxicological side effects, and high biodegradability. In particular, drying glyceride plant oils, which are rich in unsaturated fatty acids, might be promising candidates for a more sustainable approach in the discussion about plant protection and the environment. Based on this, we tested the protective and curative efficacy of an oil-in-water-emulsion preparation using drying plant oils (linseed oil, tung oil) and a semi-drying plant oil (rapeseed oil) separately and in different mixtures. Plant oils were tested in greenhouse experiments (in vivo) on green beans (*Phaseolus vulgaris* L.) against bean rust (*Uromyces appendiculatus*). We observed that a 2% oil concentration showed no or very low phytotoxic effects on green beans. Both tested drying oils showed a protective control ranging from 53–100% for linseed oil and 32–100% for tung oil. Longer time intervals of 6 days before inoculation (6dbi) were less effective than shorter intervals of 2dbi. Curative efficacies were lower with a maximum of 51% for both oils when applied 4 days past inoculation (4dpi) with the fungus. Furthermore, the results showed no systemic effects. These results underline the potential of drying plant oils as biorationals in sustainable plant protection strategies.

## 1. Introduction

It is common sense that a high level of food production is required to feed a growing world population. The “Green Revolution” [1,2] increased global food production through high-yielding crop varieties but came with certain drawbacks of the applied technologies [3]. One major input of this development is synthetic chemical plant protection products (pesticides) to protect crops against pathogens and pests [4]. However, there are increasing concerns about the environmental costs of their use [5]. Hence, efficient plant protection is required, but it must be achieved through environmentally safe measures. Many countries in Europe have developed national regulations and plans to reach these goals as it is required by the Directive 2009/128/EC [6] for the sustainable use of pesticides. For example, Germany has developed the “National Action Plan for the Sustainable Use of Plant Protection Products” to reduce the use of chemical pesticides and to expand the portfolio of non-chemical plant protection methods. The aim is to develop and disseminate environmentally friendly, natural, and biological solutions for plant protection [7]. All measures should fit into the eight principles of integrated pest management (IPM) [8]. Besides the strategies described, plant protection products should be of low risk for humans and the environment [9]. In this context, biorationals and how they could fit into IPM strategies are discussed, but a precise definition of biorationals is difficult [10,11]. Natural products like glyceride plant oils [12] are promising candidates for such a sustainable approach because of their low human toxicity, their good biodegradability [13], the chance of modifications in their composition, the miscibility with other substances or fluids [14], and their lack of environmental persistence [15]. Furthermore, in the Commission Implementing Regulation (EU) 2019/2164 [16], plant oils are listed in Annex II of substances of plant or animal origin that can be used as pesticides in organic farming. They are considered as low-risk products.

The basis for possible applications is the chemical and physical properties of the natural product considered. Glyceride plant oils consist of a glycerol molecule esterified with three long-chain fatty acids [17]. The chain length of the fatty acids, the proportion of unsaturated fatty acids, and the number, position, configuration (cis, trans) of the double bonds, and the resulting iodine value (IV) determine the oil characteristics [18]. Based on fatty acid composition and the IV, plant oils can be categorized into non-drying (IV: 75–100), semi-drying (IV: 100–150), and drying oils (IV: 150–190) (Table 1) [19]. Plant oils with a high content of unsaturated fatty acids (e.g., α-eleostearic acid in tung oil or linolenic acid in linseed oil) [19] form cross-linked films through autoxidation at ambient oxygen concentrations [20]. While linseed oil and tung oil belong to the drying category, rapeseed oil belongs to the non-drying to semi-drying category due to a lower proportion of polyunsaturated fatty acids [19,21]. The fast-drying and film-forming capacity of tung oil is mainly caused by three conjugated double bonds in the α-eleostearic acid [22,23], resulting in waterproof tissues [19]. This general behavior of drying plant oils is known [24] (pp. 29,63) and is used in the paint and cosmetic industries [20], as well as in the field of medicine [25] to cover wounds, as carriers in medicines, and in tissue regeneration where the film builds a barrier [26,27]. Despite the range of described applications and advantages, plant oils are currently not widely applied in agriculture and horticulture. In particular, drying oils have not been studied extensively in plant protection.

Clayton et al. [28] showed in the 1940s that certain plant oils, e.g., linseed oil and tung oil, reduced blue mold of tobacco caused by *Peronospora tabacina* and concluded that the fatty acid composition could be of importance for the fungicidal effect, without giving further details. In addition, Arslan [29] described that a protective application of linseed oil could reduce the infestation of *Uromyces appendiculatus* under in vivo conditions. Both studies do not discuss the possible role of film-forming capacity, which has been shown to be relevant against pests. Effects of linseed oil on the white rose scale insect [30] and tung oil against boll weevils [31] are based on this film-formation on surfaces, which coats insects and suffocates them and their eggs [19,32]. Other non-drying plant oils have been demonstrated to be effective against powdery mildew on cucurbits (*Sphaerotheca fuliginea*) [12], apples (*Podosphaera leucotricha*) [33], hop mildew (*Sphaerotheca humuli*) [34], and apple scab (*Venturia inaequalis*) [35]. Products based on rapeseed oil are currently approved as plant protection products in organic farming [36].

Besides a sufficient efficacy, the crucial point for successful integration of drying plant oils into existing plant protection strategies is the knowledge of the phytotoxic side effects and a preparation achieving good solubility or shelf life [12,37]. Here, a new oil-in-water preparation invented by Petry et al. [38] was used, which is described in detail in Section 4.1. With this preparation, a good solubility, miscibility, and shelf life of the oils could be achieved, and it could be applied with standard equipment without causing sprayer blockage. In this study, we report for the first time the fungicidal activity of this preparation.

From the literature, it could be hypothesized that the film-forming capacity of drying plant oils could act as a protective barrier on plant surfaces against a fungal infection or by interfering with the development of the pathogen after curative application. In obligate biotrophic host–parasite-interactions, the recognition processes are prerequisites for a successful establishment of the disease [39]. Furthermore, it could be assumed that the fungicidal efficacy depends on oil type, its drying category, and its fatty acid composition. Overall, our aim was to show that drying plant oils could be used in a targeted manner to interfere in the plant–pathogen-interaction, leading to a reduced disease incidence. Therefore, we investigated the effects of the drying plant oils after a protective and curative application in the host–parasite-interaction of green beans and *U. appendiculatus*. We selected linseed oil (L), tung oil (T), and rapeseed oil (R) as test plant oils according to their drying level and fatty acid composition (Table 1). Greenhouse experiments were conducted with these plant oils and mixtures of them (for detailed description see Section 4.3.3, Table 2).

## 2. Results

### 2.1. Phytotoxic Effect of Drying Plant Oils

At first, we tested possible adverse effects of the new plant oil preparations on the sensitive primary leaves of green beans using four different linseed oil (L) and rapeseed oil (R) concentrations (0.5%, 1%, 2%, and 5%) in comparison to that of untreated control (C). Visual appearances of leaf damage are presented in Figure 1.

The phytotoxic effects were further evaluated with the “Image Analysis Software (Assess 2.0)” as shown in Figure 2. Up to an oil concentration of 1%, leaf damage was less than 0.5% and similar to that of control (water). At oil concentration of 2%, leaf damage was less than that for 5%. Only an oil concentration of 5% showed a percentage of leaf area damage of more than 8% for rapeseed oil and up to 14% for linseed oil (Figure 1 and Figure 2).

### 2.2. Efficacy against Uromyces appendiculatus

A dose-response experiment was carried out to determine the most effective oil concentration. Oil concentrations from 0.5 to 5% were applied once, two days before inoculation (2dbi). Figure 3 shows the fungicidal protective effect against *U. appendiculatus* using the drying plant oil (linseed oil) and semi-drying plant oil (rapeseed oil). It should be noted that the disease incidence in water-treated control was approximately 10 colonies cm^−2^ lower than in that of other experiments. All treatments were effective and statistically significantly different from untreated control. There was no obvious effect of concentration on efficacy. For linseed oil, an efficacy of 99% was achieved even with the lowest concentration tested. In comparison, the efficacy of rapeseed oil was statistically lower, ranging from 38% to 62% (Figure 3).

After these experiments on phytotoxicity and dose-response of the fungicidal efficacy, we decided to use a 2% oil concentration of the respective oil for all further experiments. Next, possible systemic effects were evaluated. Therefore, we treated only half of the primary bean leaves (as described in Section 4.3.3) and examined the efficacy on treated and non-treated parts of the leaves. Figure 4 shows the results of the translocation experiment of the 2% linseed oil and rapeseed oil. While the efficacy on the treated part of the leaf was high for both oils (Figure 4a), it was negligible on the non-treated part of the same leaf (Figure 4b). In contrast to the former experiment, rapeseed oil reached a higher control of 90%.

In the next set of experiments, protective and curative control of the pathogen was tested in more detail, because the knowledge on timing of application is prerequisite for a successful implementation in plant protection strategies. Prolonged time intervals (days) of a single application before inoculation (6dbi, 4dbi, and 2dbi for protective control) and past inoculation (2dpi and 4dpi for curative control) were used. Furthermore, we included tung oil in these experiments as a drying oil featuring a different fatty acid composition (see Table 1), to support our idea that the fungicidal effect depends on the type of oil, its drying category, and its fatty acid composition. The single plant oil preparations (L, T, R) were compared to three commercial products (Cu, Mi, and Fu, see Section 4.3.3, Table 2 for explanation). The disease incidence (Figure 5a–e) in untreated control was comparable for all time intervals tested. All treatments showed a protective efficacy ranging from 9% (Cu, 6dbi, Figure 5a) to 100% (Fu, 4dbi, Figure 5b). While the single oils showed a protective (Figure 5a–c) inhibitory effect ranging from 32% (T, 6dbi, Figure 5a) to 66% (T, 2dbi, Figure 5c), the highest disease reduction was observed for Fu (99–100%). For the drying plant oils, efficacy increased from 6dbi to 2dbi. L ranged from 53% to 60% and T ranged from 32% to 66% (Figure 5a,c). Shorter time intervals between protective application and inoculation with the pathogen resulted in higher protection. The curative efficacy (Figure 5d,e) of the treatments was lower in comparison to that of the protective application. The highest and statistically significant disease reduction was observed for Fu with 78% (Figure 5e). All other treatments showed an inhibitory efficacy ranging from 6% (Cu, 4dpi, Figure 5e) to 51% (L and T, 4dpi, Figure 5e). Drying plant oils and Fu showed a higher inhibitory efficacy for 4dpi (Figure 5e) than 2dpi (Figure 5d) and resulted in a statistically significant difference in comparison to that of the untreated control, while R, Cu, and Mi showed a lower effect for 4dpi.

To evaluate combinatory effects of the single oils, which could be based on their respective fatty acid composition, we used four different mixtures (LT(1), LT(2), LT(3), and LTR, see Table 2). LT(1) to LT(3) represent decreasing amounts of tung oil varying from 50% to 5% in combination with linseed oil. Furthermore, a mixture of three oils (LTR) contains drying (L, T) and semi-drying oil (R) was included. Figure 6 shows the protective (2dbi and 1dbi) and curative (2dpi) fungicidal effects of seven different plant oil treatments in comparison to that of commercial products (Cu, Mi, and Fu) against *U. appendiculatus*. The disease incidence in untreated control was comparable for all scenarios. All treatments showed a protective efficacy ranging from 44–100% (Figure 6a,b) and were statistically significantly different to that of the untreated control. The highest effects (99–100%) were observed for drying plant oils, combinations of drying plant oils, and Fu. In comparison to drying oils, efficacies for R, Cu, and Mi were lower. A shorter time interval between application and inoculation of just 1 day (1dbi) resulted in similar efficacies as those observed for 2dbi. Again, the drying oils had higher efficacies compared to that of semi-drying rapeseed oil (R) and were comparable to fungicide treatment (Fu). The curative efficacies (Figure 6c) of the treatments were lower in comparison to that of a protective application, with the highest and most statistically significant disease reduction for Fu with 52%. All other treatments were less effective with disease reductions ranging from 19% (Cu) to 28% (LTR). There were no obvious differences between oil mixtures and single oil treatments, indicating that there are no combinatory effects or interactions. In this experimental series (Figure 6), higher efficacies were observed in comparison to those seen in the earlier experiments (Figure 5), demonstrating that fungicidal effects may depend on factors that could not be controlled in the study (e.g., slight differences in environmental conditions, physiological status of plants). In all experiments, the protective control of *U. appendiculatus* by drying oils ranged between 32% and 100%.

## 3. Discussion

In this study, oil concentrations of up to 2% led to no or very low adverse side effects in sensitive primary leaves of green beans. In our studies, we decided to use 2% oil instead of lower concentrations to safeguard fungicidal efficacy throughout the different experimental series. A 2% oil concentration resulted in significant control of bean rust in multiple trials, drying oils (L,T) were more effective than the semi-drying oil (R), but oil mixtures show no beneficial effect. A protective application (up to 6dbi) showed higher efficacies than curative treatment. The observed effects were not systemic.

Fungicidal activity of plant derived triglyceride oils on fungal diseases like hop powdery mildew [34] or blue mold of tobacco [28] are known, but the utilization in agricultural practice is still limited, with one reason for limited use being that preparations used could lead to blockages in the sprayer nozzles [30]. Our studies aimed to test if an innovative preparation of drying oils could be a promising tool in an environmentally friendly plant protection.

In general, it could be shown that a protective application of the drying linseed oil (L) and tung oil (T) from 2 up to 6 days before inoculation resulted in an effective control of bean rust. These results are in accordance with research from Arslan [29], although a much shorter time interval (2 hours before inoculation) was used. These findings support the results shown in Figure 6. The infestation could be reduced by up to 100% by a 2% concentration of a protectively-applied drying linseed oil-in-water-emulsion 48 h before inoculation. Clayton et al. [28] showed a fungicidal effect of linseed oil and tung oil against blue mold of tobacco plants. They applied different plant oils several times using concentrations of 1% and 2%. They concluded that the best results could be achieved using 2% linseed or tung oil and related these results to the high content of linolenic acid or α-eleostearic acid, respectively (Table 1).

Contrary to our findings, Arslan [29] showed that curative treatment with linseed oil was ineffective against *U. appendiculatus* when applied 24 h after inoculation. In our study it could be shown that a curative efficacy against bean rust can be achieved up to 4 days after inoculation (Figure 5). The different levels of efficacy in the cited studies compared to that in our results could be explained by the different preparation of the oil-in-water-emulsion used. This finding shows the importance of the preparation and formulation of drying oils for their efficacy against pathogens and pests, which is reported by Ah Chee et al. [12] as well.

Northover and Timmer [42] (pp. 512–526) reported that so called “Horticultural Mineral Oils” and glyceride plant oils have a short protective activity of up to three days only. This finding could partly be confirmed by our results showing that shorter protective time intervals lead to a higher efficacy (Figure 5). However, it should be mentioned that the efficacy could vary depending on further experimental factors (e.g., cultivation, pathogen virulence), which cannot be controlled completely (Figure 5 and Figure 6).

Drying oils are characterized by their high content of unsaturated fatty acids [18,19], which are responsible for the films forming [23] on surfaces through rapid autoxidative drying at ambient oxygen concentration [20,22]. These properties of the drying plant oils support the presented results, but could be further investigated by detailed microscopic studies. The drying effect and the resulting film formation on plant surfaces may interfere with fungal infection and growth by masking infection sites, forming an additional barrier, and impairing plant–pathogen-interactions. This idea is supported by the findings of Chen and Ko [43], who discovered that linseed oil reduced the germ tube length of peanut rust and suppressed the appressorium formation. Mendgen [39] demonstrated for *Uromyces phaseoli* (syn.: *U. appendiculatus*) that only when the germinal tube differentiates into an appressorium, will the infection be successful. Furthermore, it is known that tung oil can have a gelatinous consistency [19], and due to this characteristic it is used in the production of waterproof fabric and paper. It could be concluded that the efficacy of drying oils against pathogens can partly be explained by their film-forming capacity, which builds a layer on the plant surface as an additional physical and chemical barrier, impairing the infection process. Whether this effect is based on fungistatic [44] or fungicidal activity [34] has been discussed and remains open for debate. In summary, we showed that linseed oil and tung oil can provide a protective control of bean rust when applied 2 or 4 days before inoculation. The efficacy decreases with a prolonged time interval of 6dbi. A curative control up to 4dpi was also demonstrated.

Our trials demonstrated good fungicidal efficacies of two drying oils preparations based on linseed oil and tung oil. As natural products with low ecotoxicity and good biodegradability, drying oils are promising candidates as biorationals in sustainable and environmentally friendly plant protection strategies, which would also be applicable in organic farming. In further studies, more pathogens will be tested and mode of action should be investigated in detail.

## 4. Materials and Methods

### 4.1. Plant Oil Preparations and Application

The plant oils used were made from seeds of flax plants (*Linum usitatissimum* L.), nuts of tung trees (*Aleurites fordii*, *Vernicia montana*, *Vernicia fordii*), and seeds of rape plants (*Brassica napus* L.). The oils contained triglycerides with varying combinations of fatty acids. Linseed oil and tung oil contained high amounts of 18:3 fatty acids, which are responsible for the drying character of these oils (Table 1). Two drying plant oils, linseed oil (L) and tung oil (T), and the semi-drying rapeseed oil (R) were used in the experiments. The processing and formulation of emulsified drying plant oils were based on a patent by Petry et al. [38]. “PETRYmade” (Meckenheim, Germany) produced the oils used in this study.

The basis of the oil preparations used in this study were emulsions of the tested plant oils in water. Under exclusion of oxygen, the single plant oils were slowly heated up to 230–280 °C to reach an acid number below 10 [19,21]. The exact temperature rise, duration, and the final temperature of the boiling processes were dependent on the type of oil. The resulting resins were cooled and an emulsifying agent of less than 5% was added to make the resins water-soluble. The stock emulsions of the plant oils contained 50% water and 50% of the respective pure plant oil including the emulsifying agent. Further details of the preparation can be found in the patent description by Petry et al. [38]. The stock emulsion was then diluted with water to the desired test concentrations (Table 2).

Plant oil application was performed once with hand sprayers on the abaxial and adaxial side of primary leaves of uniformly developed three-week-old green bean seedlings until runoff. Control (water) and commercial products were applied in the same way.

### 4.2. Phytotoxic Effects of Drying Plant Oils

Primary leaves of bean plants (Phaseolus vulgaris L. cv. ‘Saxa’) were treated with four concentrations (0.5%, 1%, 2%, and 5%) of linseed oil and rapeseed oil (see Section 4.3.3, Table 2). Phytotoxic effects in the form of leaf damage (%) were assessed by visual inspection and compared to a control (plants treated with water). Furthermore, twelve days after application, photographs were taken of the leaves, which were evaluated using the “Image Analysis Software (Assess 2.0)”.

### 4.3. Efficacy against Uromyces appendiculatus

#### 4.3.1. Test Pathogen, Inoculum Preparation, and Inoculation

Uredospores of *Uromyces appendiculatus* were provided by the Institute of Crop Sciences and Resource Conservation, University of Bonn, Germany. The uredospores were gently dislodged with a brush from the surface of rust-infected leaves and were stored in a freezer at −18 °C until use. For inoculation, the uredospore concentration was determined with a hemocytometer and adjusted to a spore density of 6 × 10^4^ spores mL^−1^, resulting in a sufficient and reliable number of rust pustules in the control. The spores were suspended in water by adding 0.01% of the wetting agent Tween® 20 (Merck Group, Darmstadt, Germany). Following the procedure of Bassanezi et al. [45], primary leaves were sprayed with spore suspension until runoff by hand sprayers. Inoculated potted plants were placed in trays with water and covered with polyethylene covers to increase humidity and to facilitate infection. The covered plants were kept at 19 ± 1 °C for 48 h in darkness. Afterwards, polyethylene covers were removed and the inoculated plants were further cultivated under greenhouse conditions. Disease incidence was determined 10–14 days after inoculation, when rust pustules were visible on plants. For the disease incidence, the number of visible fungal colonies per cm^2^ leaf area was counted on four random sites per plant using a circle screen, with five replicates per treatment, and the efficacy was calculated according to Abbott [41].

#### 4.3.2. Plant Material and Growth Conditions

Green beans (*Phaseolus vulgaris* L. cv. ‘Saxa’) were grown in pots of 10 cm diameter in commercial growing substrate ED73 (containing white peat, raised-bog peat, and natural clay). Pots were placed on low-tide-high-tide tables with a daily irrigation. They were cultivated for three weeks under greenhouse conditions (18 ± 1 °C during the night and 20 ± 2 °C during the day, 12 h of light per day).

#### 4.3.3. Trial Descriptions

Dose-response: For assessment of the dose-response of efficacy, four concentrations (0.5%, 1%, 2%, and 5%) of linseed and rapeseed oil were used (Table 2). Plant oils were applied 2dbi as described in Section 4.1.

Translocation: For translocation of the effects in the same leaf, only half of the primary bean leaves were treated 2dbi. The distal half of the leaf was immersed for a few seconds in oil preparation or water as control. After application, plants were carefully handled to avoid runoff of the oil onto the untreated half of the leaf.

Prolonged protective and curative efficacies: In this experiment, tung oil (T) was included as an alternative drying oil. In total, 2% linseed oil (L), tung oil (T), and rapeseed oil (R) were used and compared to three commercial products: Funguran® progress (Cu) (a.i.: 537 g/kg copper hydroxide, Biofa AG, Münsingen, Germany), Micula® (Mi) (a.i.: 85% emulsified rapeseed oil, Biofa AG, Münsingen, Germany), and Flint® (Fu) (a.i.: 500 g/kg Trifloxystrobin, Bayer CropScience, Mohnheim, Germany). Funguran® progress and Micula® were chosen because they are authorized products in organic farming [36]. The three reference products were used following manufacturer recommendations. Application intervals of six days (6dbi), four days (4dbi), and two days (2dbi) before inoculation were used for protective effects. Application intervals of two days (2dpi) and four days (4dpi) past inoculation were tested to evaluate curative efficacy.

Combinatory effects of oil mixtures: To test possible combinatory effects of oil mixtures, three different ratios of linseed oil and tung oil (LT(1), LT(2), and LT(3)), and a mixture of all three oils (LTR) were used and compared to single oils and commercial products. Details on the ratios are given in Table 2. Oils and commercial products were applied 2dbi, 1dbi, and 2dpi.

Application (except for translocation experiment), pathogen inoculation, and disease assessment were carried out as described in Section 4.1 and Section 4.3.1

### 4.4. Statistical Analysis

The experiments were conducted in a completely randomized design, with five replications per treatment (*n* = 5). In all Figures, data are presented as mean and standard deviation (X̅ ± σ); number of replicates (*n*) is given. Analysis of variance (ANOVA) was used to compare means of treatments. Under given normal distribution (Kolmogorov–Smirnov test) and homoscedasticity (Levene test) a Tukey-HSD test was used as the post hoc procedure to determine homogeneous subgroups at a *p* value of *p* ≤ 0.05. Data that did not meet homogeneity of variance or normal distribution were analyzed using the Games-Howell test at a *p* value of *p* ≤ 0.05. All statistical analyses were performed using IBM SPSS 25.0 software.

## 5. Conclusions

The presented study provides information on the efficacy of two drying plant oils (linseed oil and tung oil) against the pathogen *Uromyces appendiculatus*. The efficacy can partly be explained by the film-forming capacity of drying plant oils. A thin film on the plant surface can provide an additional barrier against fungal infection. Host–parasite interactions could be impaired. Thus, a protective and curative fungicidal activity can be achieved. These results underline the potential of drying plant oils as biorationals for organic farming as non-toxic and environmentally friendly products.

## 6. Patent

Petry, M.; Pude, R.; Kraska, T. Verfahren zur Herstellung eines Pflanzenbehandlungsmittels und Verfahren zur Behandlung von Pflanzen. DE102017000110A1; EP3568014A1, 1 October 2017. Available online: https://worldwide.espacenet.com/patent/search?q=pn%3DEP3568014A1 (accessed on 03 November 2020).

## Figures and Tables

**Figure 1 plants-10-00143-f001:**
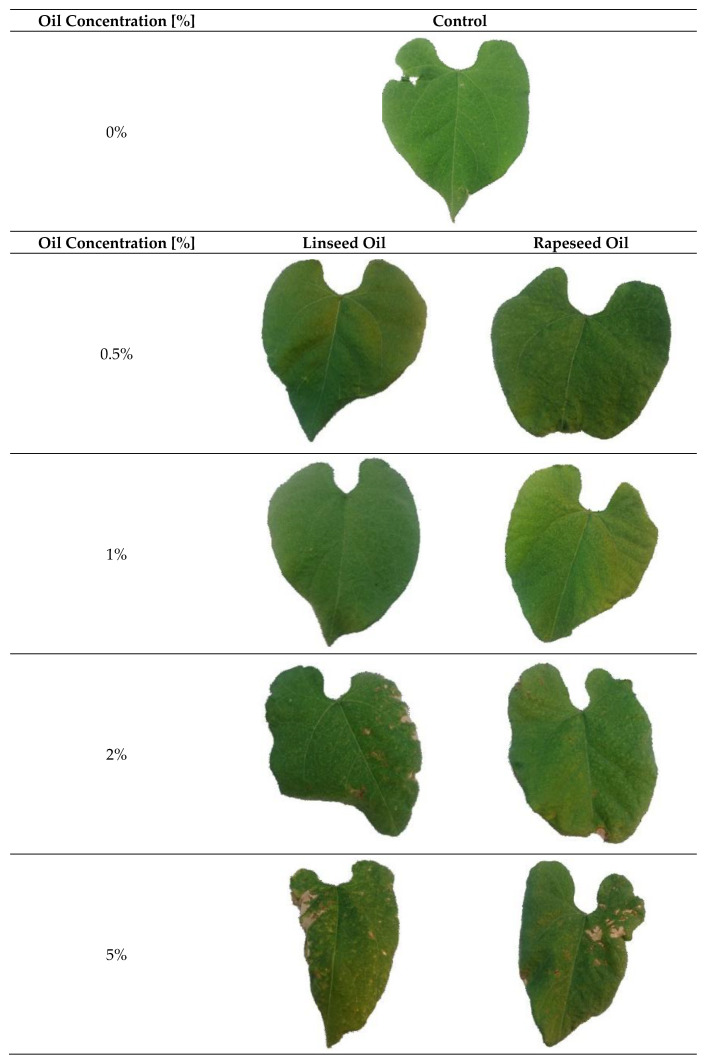
Visible leaf damage of primary bean leaves treated with different oil concentrations of linseed oil and rapeseed oil in comparison to that of untreated control.

**Figure 2 plants-10-00143-f002:**
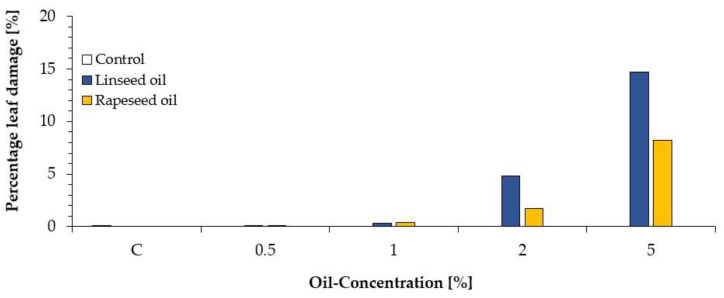
Phytotoxic effect (leaf damage in %) depending on oil concentration in comparison to an untreated control (white).

**Figure 3 plants-10-00143-f003:**
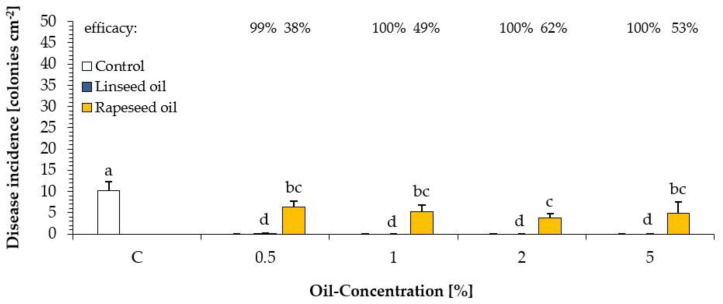
Disease incidence of bean rust (colonies cm^−2^ leaf) on leaves of *Phaseolus vulgaris* after a protective application (2dbi) of four different oil concentrations of the drying linseed oil (blue) and semi-drying rapeseed oil (yellow) in comparison to that of an untreated control (white). Different letters above the columns indicate statistically significant differences at *p* < 0.05 according to the Games-Howell test. Percent values above represent the reduction of bean rust infestation in % in comparison to that of untreated control (according to Abbott [41]). *n* = 5.

**Figure 4 plants-10-00143-f004:**
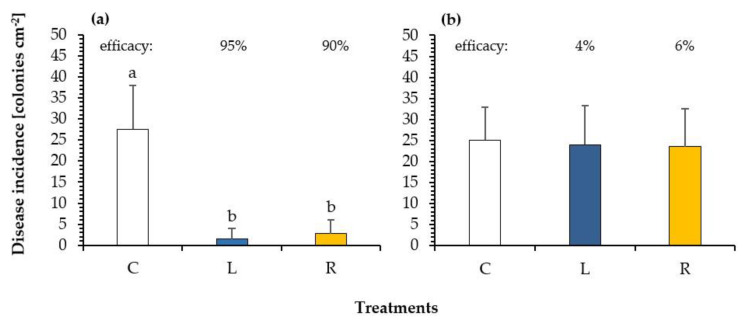
Disease incidence of bean rust (colonies cm^−2^ leaf) on (**a**) the half of primary leaves of *Phaseolus vulgaris* after a protective treatment (2dbi) with 2% drying linseed oil (blue) and 2% semi-drying rapeseed oil (yellow) in comparison to that of an untreated control (white) and (**b**) on the other non-treated half of the same leaves. Different letters above the columns indicate statistically significant differences at *p* < 0.05 according to the Games-Howell test. Percent values above represent the reduction of bean rust infestation in % in comparison to that of untreated control (according to Abbott [41]). *n* = 5.

**Figure 5 plants-10-00143-f005:**
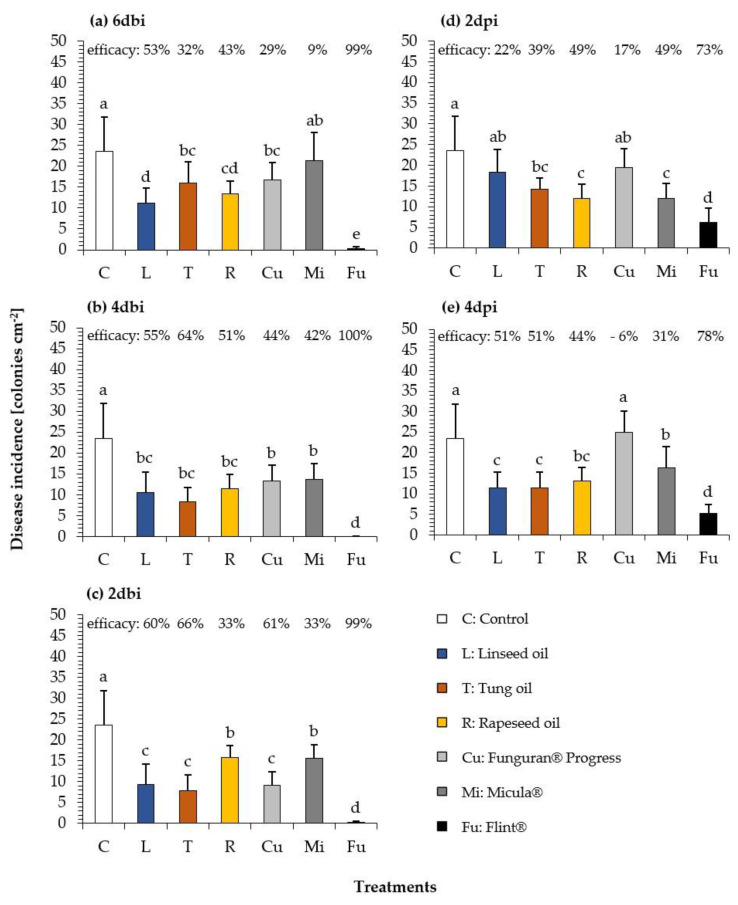
Disease incidence of bean rust (colonies cm^−^² leaf) on leaves of *Phaseolus vulgaris* after (**a**) protective application of 6dbi, (**b**) 4dbi, and (**c**) 2dbi as well as (**d**) curative application of 2dpi, and (**e**) 4dpi of three different oil treatments (L, T, and R) and three commercial products (Cu, Mi, and Fu) in comparison to that of untreated control (C). Used concentrations: 2% L, T, and R; Cu (537 g/kg Cu(OH)_2_), Mi (85% emulsified rapeseed oil); Fu (500 g/kg Trifloxystrobin). Different letters above columns indicate statistically significant differences at *p* < 0.05 according to the Games-Howell test. Percent values above represent the reduction of bean rust infestation in % in comparison to that of untreated control (according to Abbott [41]). *n* = 5.

**Figure 6 plants-10-00143-f006:**
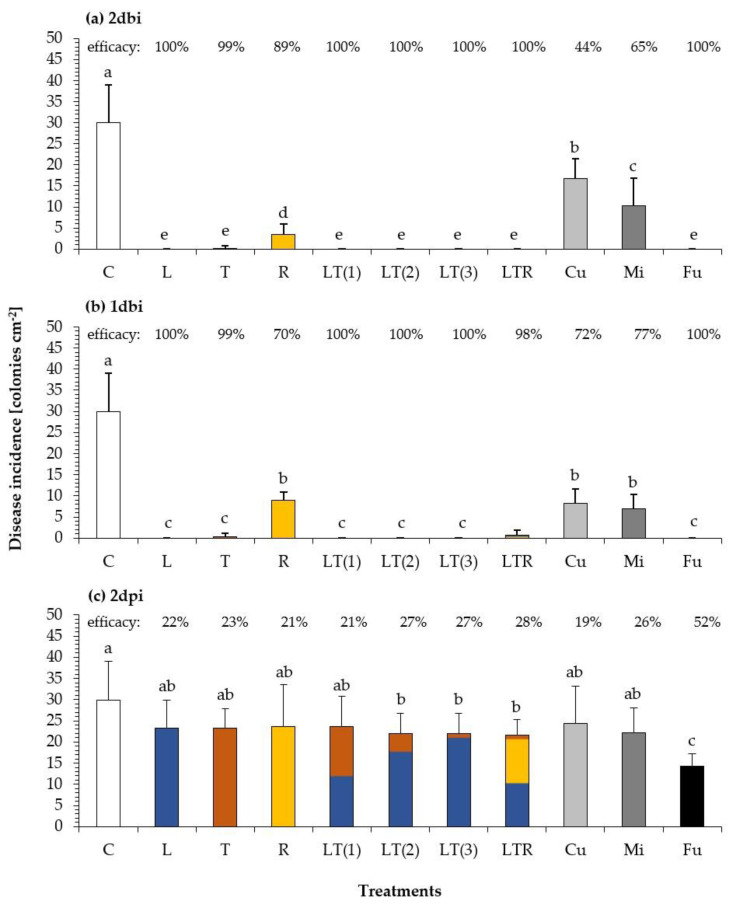
Disease incidence of bean rust colonies cm^−^² on leaves of *Phaseolus vulgaris* after (**a**) protective application of 2dbi, (**b**) 1dbi, and (**c**) curative application of 2dpi of three single oils (L, T, and R), four oil mixtures (LT(1)—50:50, LT(2)—80:20, LT(3)—95:5, and LTR—47.5:5.0:47:5), and three commercial products (Cu, Mi, and Fu) in comparison to that of untreated control (C). Used concentrations: 2% for all tested oil combinations; Cu (537 g/kg Cu(OH)_2_), Mi (85% emulsified rapeseed oil); Fu (500 g/kg Trifloxystrobin). Different letters above indicate mean values significantly different at *p* ≤ 0.05 according to the Games-Howell test. Numbers above represent the inhibitory efficacy on bean rust infestation in % in relation to untreated control (according to Abbott [41]). *n* = 5.

**Table 1 plants-10-00143-t001:** Characterization of the plant oils used in this study, listing their main fatty acids (only fatty acids with a content >10% are listed), iodine values (IV), and drying categories.

Oil Characteristics	Fatty Acid Content (%) of			Reference
	Linseed Oil ^1^	Tung Oil ^1^	Rapeseed Oil ^1^	
**Fatty Acid *** Stearic Acid, 18:0 Oleic Acid, 18:1 (9c) Linoleic Acid, 18:2 (9c,12c) Linolenic Acid, 18:3 (undefined) ^2^ α-Eleostearic Acid, 18:3 (9c,11t,13t)	2–16 14–39 7–25 35–66 -	1.3–2.7 8–14.9 10.9–11.5 - 63.8–79.7	1.1–2.5 33–67 16–25 6–14 -	[21,40]
**Iodine Value (IV) ^3^**	155–205 165–190	147–175 147–172	110–126 94–120	[21] [19]
**Drying Category**	drying	drying	non-drying to semi-drying	[19]

*—The number after the fatty acids represents the number of C-atoms/double bonds (position, configuration). ^1^—The fatty acid composition refers to different plant species, specifically, the fatty acid composition of tung oil is a summary of the following species (*Aleurites fordi*, *Vernicia montana*, *Vernicia fordii*); linseed oil (*Linum usitatissimum*); rapeseed oil (*Brassica napus* L.) (low in erucic acid, canola)—compositions depend on the species. ^2^ Summarizing α-Linolenic acid (9c, 12c, 15c) and γ-Linolenic acid (6c, 9c, 12c). ^3^—Definition of the iodine value (IV) (according to Roth and Kormann [19]): The iodine value indicates how much halogen, expressed as a percentage of iodine, a fat/fatty acid can add. Depending on the number of double bonds, the unsaturated fatty acids add 2, 4, 6, or more atoms of iodine (the corresponding fatty acid halide is formed), while the saturated fatty acids do not add iodine. Configuration: c—cis, t—trans configuration.

**Table 2 plants-10-00143-t002:** Information about the tested plant oils and treatments, abbreviations for the treatments used in the text, ratio used in oil mixtures, and information on oil concentrations and active ingredients of commercial products.

**Plant Oil**	**Abbreviation**	**Ratio**	**Total Oil Concentration**
Linseed	L L	- -	2% 0.5%, 1%, 2%, 5% ^1^
Tung	T	-	2%
Rapeseed	R R	- -	2% 0.5%, 1%, 2%, 5% ^1^
Linseed: Tung	LT(1)	50:50	2%
Linseed: Tung	LT(2)	80:20	2%
Linseed: Tung	LT(3)	95:5	2%
Linseed: Tung: Rapeseed	LTR	47.5:5:47.5	2%
**Commercial Products**	**Abbreviation**		**Active Ingredient**
Funguran® Progress	Cu		537 g/kg Cu(OH) ^2^
Micula®	Mi		85% emulsified rapeseed oil
Flint®	Fu		500 g/kg Trifloxystrobin
Control	C		water

^1^ Final oil content in the oil-water-emulsion. ^2^ Concentrations used for phytotoxicity and dose-response experiments.

## Data Availability

The data presented in this study are available on request from the corresponding author.

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
