# Peer review of "Fungicidal Efficacy of Drying Plant Oils in Green Beans against Bean Rust (Uromyces appendiculatus)"

_plants, 2021, doi:10.3390/plants10010143_

Round 1
Reviewer 1 Report
The work is complete and well constructed even if the results do not provide univocal data, as described by the authors themselves
1) The bioactivity of two drying vegetable oils (linseed oil, tung oil) and a semi-drying vegetable oil (rapeseed oil), separately and in different mixtures, was tested in vivo on green beans (Phaseolus vulgaris L.) against the rust of beans (Uromyces appendiculatus). In particular, linseed oil and tung oil applied 2 days before the pathogen inoculation showed protective control, with no or very low phytotoxic effects. The results lead to hypothesize the possible use of dried vegetable oils in sustainable plant protection strategies.
2) The experimental setting and the evaluation of the results are conducted following the appropriate scientific methodology, although the results show a certain degree of variability, probably due to the complex nature of the used oils, as reported in Tab. 1. This limitation is however also highlighted by the authors themselves.
3) I also believe that the work is well structured and does not require particular improvements.
Author Response
Dear Reviewer,
thank you very much for your positve response.
You are completey right and we agree to your statements on "univovcal" and "limitations". Data for the effects of natural products or biorationals are often "not univocal" (as you have mentioned it). They are depending on environmental factors and experimental conditions as we have stated in the manuscript. We intended to present the range of effects and compared it to commercial standards.
Thank you very much for taking the time to review or study. We really appreciate it.
Regards,
Thorsten Kraska
Reviewer 2 Report
The article is interesting. Several mistakes observed. Question how did the authors evaluated disease incidence? What scale?
I want to clarify the applications were 1 DPI, 2 DPI and 2 DPI or 4 DPI on same plants? Were the applications once a time or several times a week? Please clarify this in methods.

Author Response
Dear Reviewer
thank you very much for reviewing our manuscript and your valuable comments. Wwe really appreciate that you have taken your time to suggest corrections in our text. We considered them all very carefully and made changes. Changes have been made and highlighted, except for
1 -suggestion "was not very important" (page 6). I kept the term "negligible" here. We want to clearly state that the effect was very low. We would nlot like to make a reference on if this no-effect is also not important
2. You suggest to use the term "intense". Here I rephrased the sentence (page 9)
3 - polyethylene: no mispelling (page 12)
Reviewer: I want to clarify the applications were 1 DPI, 2 DPI and 2 DPI or 4 DPI on same plants?
Answer: Plants were treated once. So for each time interb´val new plants were used. Otherwise one could not separate them into the different treatment groups. The application was before (protective, dbi) or past inoculation (curative, dpi) as stated in the text.
Reviewer: Were the applications once a time or several times a week?
Answer: They were applied "once" as stated in Material & Methods (page 11, highlighted).
Thank you again for the time to read and review the manuscript. I hope that the made changes meet your comments.
Regards,
Thorsten Kraska
Reviewer: Question how did the authors evaluated disease incidence? What scale?
Answer: I have highlighted the description in the text. I have rephrased the description slightly to make it clearer.